# Correlations between Kidney and Heart Function Bioindicators and the Expressions of Toll-Like, ACE2, and NRP-1 Receptors in COVID-19

**DOI:** 10.3390/vaccines10071106

**Published:** 2022-07-11

**Authors:** Rabab Hussain Sultan, Basem H. Elesawy, Tarek M. Ali, Maged Abdallah, Hebatallah Hany Assal, Amr E. Ahmed, Osama M. Ahmed

**Affiliations:** 1Biotechnology and Life Sciences Department, Faculty of Postgraduate Studies for Advanced Sciences, Beni-Suef University, Beni-Suef 62511, Egypt; amreahmed@psas.bsu.edu.eg; 2Department of Pathology, College of Medicine, Taif University, P.O. Box 11099, Taif 21944, Saudi Arabia; b.elesawy@tu.edu.sa; 3Department of Physiology, College of Medicine, Taif University, P.O. Box 11099, Taif 21944, Saudi Arabia; t.hussien@tu.edu.sa; 4Department of Anesthesia and Intensive Care, Faculty of Medicine, Cairo University, Cairo 11562, Egypt; maged.salah@kasralainy.edu.eg; 5Department of Chest Medicine, Faculty of Medicine, Cairo University, Cairo 11562, Egypt; hebatallah.assal@kasralainy.edu.eg; 6Physiology Division, Department of Zoology, Faculty of Science, Beni-Suef University, Beni-Suef 62511, Egypt; osama.ahmed@science.bsu.edu.eg

**Keywords:** renal function, cardiac enzymes, ACE2, NRP-1, TLRs, COVID-19

## Abstract

Background: COVID-19 impacts the cardiovascular system resulting in myocardial damage, and also affects the kidneys leading to renal dysfunction. This effect is mostly through the binding with angiotensin-converting enzyme 2 (ACE2) and Neuropilin-1 (NRP-l) receptors. Toll-Like Receptors (TLRs) typically combine with microbial pathogens and provoke an inflammatory response. Aim: This work aims to compare the changes in kidney and heart function bioindicators and expressions of TLRs (TLR2 and TLR2) as well as ACE2 and NRP-l receptors in moderate and severe COVID-19 patients. The correlations between kidney and heart function bioindicators and expressions of these receptors are also studied. Patients and Methods: In this study, 50 healthy control and 100 COVID-19 patients (55 males and 45 females) were enrolled. According to WHO guidelines, these participants were divided into severe (50 cases) and moderate (50 cases). Serum creatinine, blood urea, CK-MB, LDH, and Troponin I were estimated. We measured the gene expression for Toll-Like Receptors (TLR2 and TLR4), ACE2, and NRP-1 in the blood samples using quantitative real-time PCR (qRT-PCR). Results: In comparison with the healthy group, all patients exhibited a significant elevation in serum creatinine, urea, cardiac enzymes (CK-MB and LDH), and CRP. Serum Troponin I level was significantly increased in severe COVID-19 patients. Furthermore, all studied patients revealed a significant elevation in the expression levels of TLR2, TLR4, ACE2, and NRP-1 mRNA. In all patients, CK-MB, ACE2, and NRP-1 mRNA expression levels were positively correlated with both TLR2 and TLR4 expression levels. Moreover, serum creatinine and urea levels were positively correlated with both TLR2 and TLR 4 expression levels in the severe group only. In the moderate group, serum CK-MB activity and Troponin I level had a significant positive correlation with both NRP-1 and ACE2 expression levels, while serum urea level and LDH activity had a significant positive correlation with NRP-1 only. In severe patients, the increases in serum creatinine, urea, CK-MB, and LDH were significantly associated with the elevations in both ACE2 and NRP-1 expression levels, whereas serum Troponin I level had a positive direct relationship with NRP-1 only. Conclusions: Our study concluded that expression levels for TLR2, TLR4, ACE2, and NRP-1 mRNA in both severe and moderate patients were positively correlated with renal biomarkers and cardiac enzymes. Innate immune markers can be important because they correlate with the severity of illness in COVID-19.

## 1. Introduction

Coronaviruses are described as a large family of enveloped RNA viruses with a single positive strand. They are capable of infecting humans and many animal species. Based on their pathogenicity, human coronaviruses can be categorized into many types.

Highly pathogenic types include severe acute respiratory syndrome coronavirus (SARS-CoV), middle east respiratory syndrome coronavirus (MERS-CoV), and the new severe acute respiratory syndrome coronavirus 2 (SARS-CoV-2) [1]. SARS-CoV-2 has led to the severest pandemic of this century with coronavirus disease 2019 (COVID-19). With the world’s second year of the coronavirus pandemic, governments are still struggling to immunize their populations to the herald immunity levels.

On 5 November 2021, the total confirmed cases reached about 248,467,363, with 5,027,128 deaths due to COVID-19 worldwide [2]. Originally recognized as a respiratory disease, COVID-19 interacts with the cardiovascular system and causes damage to the heart muscle, and results in cardiac and endothelial dysfunction principally via the angiotensin-converting enzyme 2 (ACE2) receptor [3]. Cell entry receptors are certainly the key factors that determine the tropism and influence the severity of infection of a specific virus. Likewise, a high incidence of viral mutations can enable these viruses to alter their specificity or binding affinity to a specific receptor [4]. ACE2 was originally identified in 2000 as a counterpart of the ACE receptor. Situated on the X chromosome and accurately mapping to chromosomal location Xp22, comprises 18 exons and 20 introns, generating 6 variants by alternative splicing [5,6].

The different tissue tropism between SARS-CoV and SARS-CoV-2 raised the possibility of additional host factors being involved. SARS-CoV-2 advanced protein comprises a cleavage site for protease furin which is not found in SARS-CoV [7]. The release of inflammatory cytokines, such as interferons (IFNs) caused by SARS-CoV-2, can increase the expression of ACE2 and potentiate the infection [8]. Neuropilin-1 (NRP-1), being a member of the signaling protein family, was shown to serve as an entry factor that potentiates the SARS-CoV-2 infectivity in vitro. This cell surface receptor has a disseminated expression and is important in viral entry, immune function, and many other actions [9]. NRP-1 is involved in SARS-CoV-2 infection, including possible spread through the olfactory bulb and into the central nervous system. Increased NRP-1 mRNA expression in the lungs of severe Coronavirus Disease 2019 (COVID-19) has been noticed. Up-regulation of NRP-1 protein in diabetic kidney cells alludes to its importance in severe COVID-19. Involvement of NRP-1 in immune function is convincing, given the role of an exaggerated immune response in disease severity and deaths due to COVID-19. NRP-1 has been suggested to be an immune checkpoint of T cell memory [9]. Cantuti-Castelvetri et al. illustrated that NRP-1, which is known to bind furin-cleaved substrates, increases the infectivity of SARS-CoV-2 [10]. Viral invasion actuates the host immune system inducing the creation of a large amount of cytokines and interferons to rule out pathogens. Taking the exception of viral DNA/RNA, the viral proteins are also considered targets of the model recognition receptor. Balance of the immune system is critical to invading pathogen recognition, killing, and elimination. One of the important key regulators of the innate immune system is the Toll-like receptors (TLRs). They help to identify the self- and non-self-molecule and eventually eliminate the non-self. Endosomal TLRs, mainly TLR3, TLR7, TLR8, and membrane-bound TLR4, have a role in the induction of cytokine storms [11]. 

Membrane-linked receptors (TLRs 1, 2, 4, 6, and 10) are important in viral protein recognition [12]. Distinct TLRs play a protective and harmful role for a specific virus. The real mechanistic vision of the pathogenicity of SARS-CoV-2 remains uncertain. This is attributable to the lack of knowledge about why the virus chose humans as its primary host and in what way the virus can escape the innate immune system in humans. In particular, the interactions between human TLRs and viral antigens, and the mechanism of cytokine storms that affect several human organs, are not mostly known. However, the pathophysiology of COVID-19 usually includes the invasion of the virus into the pulmonary alveoli, usually via the respiratory tract, primarily by way of respiratory droplets, through the airways [12]. 

The spike protein, a viral glycoprotein on its capsid, binds to the ACE2 receptor; then, the RNA genome comes into the host cell through the receptor [13]. After being inside the host cell, viral RNA replicates are formed from mRNA, resulting in the quick reproduction of viral RNA and other required structural proteins. Then again, the viral antigens interact with the host’s immune cells, and this interaction initiates pro-inflammatory reactions such as vasodilation, increased capillary permeability, and the gathering of humoral factors [14]. Both NRP-1 and ACE2 are expressed in the kidney and are linked to various renal diseases [15]. All of these factors cause the impedance of gas exchange and dyspnea. The exact cause of SARS-CoV-2 and the character of each constituent of the innate and adaptive immune systems are still unidentified [16]. The deficiency of a comprehensive consideration of the pathogenic and immunologic characteristics of the virus made the situation even more alarming to humankind [14].

The heart is a retrosternal muscular organ situated in the center of the chest. Cardiac enzymes have been used since the mid-20th century in assessing patients with suspected acute myocardial infarction (MI). More sensitive and specific biomarkers have replaced them. Troponins are the most recognizable and important biomarker used in the diagnosis of acute myocardial ischemia in modern medicine. Most patients with an acute MI have a rise in troponins within 2 to 3 h of arrival at the emergency department, compared with 6 to 12 h with creatine kinase [17]. Lactate dehydrogenase (LDH) emerged as another potential biomarker for detecting myocardial ischemia. LDH increases in the blood 6 to 12 h after an acute MI, peaks within 24 to 72 h, and normalizes within 8 to 14 days [18]. Since it is not a specific marker for cardiac myocytes, and its levels can also increase in many other conditions, LDH is no longer used in the diagnosis of myocardial infarction. Nowadays, the only usage for LDH in the evaluation of acute MI is to differentiate acute from subacute MI in patients with elevated troponin levels and normal creatine kinase (CK) and creatine kinase-MB (CK-MB) levels [19]. The kidneys play a vital role in the excretion of waste products and toxins such as urea, creatinine, and uric acid, regulation of extracellular fluid volume, serum osmolality, and electrolyte concentrations, as well as the production of hormones such as erythropoietin and 1,25 dihydroxy vitamin D and renin. The inconvenience associated with the use of exogenous markers, specifically that the testing has to be performed in specialized centers, and the difficulty to assay these substances has encouraged the use of endogenous markers. The most commonly used endogenous marker for the assessment of glomerular function is creatinine. Urea or BUN is a nitrogen-containing compound formed in the liver as the end-product of protein metabolism and the urea cycle [20]. Therefore, this study aims to evaluate the correlations between mRNA expression levels of TLRs (2 and 4), ACE2 and NRP-1 receptors, and cardiac and renal dysfunction biomarkers in COVID-19.

## 2. Patients and Methods

### 2.1. Study Population

We received laboratory-confirmed medical records and edited data of COVID-19 inpatients admitted to the National Health Commission in the period from December 2019 to January 2020. The data deadline for the study was March 2021. All the participants were isolated at Misr International Hospital, Cairo, Egypt, for the period from March 2021 to April 2021. The study was approved by the ethics committee of Misr International Hospital, Cairo, Egypt, and the institutional review board of the Ministry of Health, Cairo, Egypt (No. 3—2021/19). Severe COVID-19 patients are defined as those having hypoxemia (≤93% percutaneous oxygen saturation (SpO2) or ≥30/min respiratory rate on room air who are on the high-flow nasal cannula or noninvasive mechanical ventilation). Patients not fulfilling the above criteria were weighed as moderate. Informed printed consent was taken from all participants after the ethical committee of the institutional review board permitted this research. Cases of COVID-19 were confirmed by the presence of positive results for high-throughput sequencing or real-time reverse transcriptase-polymerase chain reaction (RT-PCR) testing of swab samples in the nose and throat [21].

### 2.2. Patients

Only cases identified as COVID-19 positive by RT-PCR in the laboratory were involved in the study. Based on how severe their symptoms were, confirmed cases of COVID-19 were divided into two groups (50 patients each) [22].

Group1 included patients with moderate symptoms and infected by a coronavirus.

Group2 included patients with severe symptoms, infected by a coronavirus, and admitted to the intensive care unit (ICU).

### 2.3. Inclusion and Exclusion Criteria

Healthy individuals were chosen as COVID-19-free volunteers. All patients were recruited between 20 and 70 years old and were diagnosed as having COVID-19 according to their PCR and chest CT, as identified by WHO 2020. Key exclusion criteria included hypertensive patients treated with ACE2 inhibitors, pre-existing respiratory disorder, kidney or liver failure, thyroid dysfunction, autoimmune disorders, cerebrovascular diseases, heart diseases, and pregnant and lactating women. In addition, patients who receive immunomodulatory drugs or those who have medical conditions such as other infections, malignant tumors, or alcohol abuse were also excluded. 

### 2.4. The Demographic Data

Anthropometric variables such as gender and body mass index (BMI) were obtained. 

### 2.5. Blood Samples

Serum samples were rapidly separated, aliquoted, and stored at −40 °C until the biochemical measurements. Serum CK-MB, C-reactive protein (CRP), and Troponin I were determined by a standard sandwich enzyme-linked immune-sorbent assay (ELISA) kit from R&D Systems (USA) as guided by the manufacturer. Gene expression of ACE2 mRNA, NRP-1 mRNA, TLR2, and TLR4 were spotted in the healthy control, moderate and severe groups in blood samples using quantitative real-time PCR (qRT-PCR). Gene sequence (5’-3’) procedures were performed, conferring to the kit instructions provided in the laboratory assay.

### 2.6. Laboratory Assays

#### 2.6.1. Detection of Serum Biomarkers of Kidney and Heart Functions

Serum creatinine levels were determined using reagent kits bought from Diamond Diagnostics (Cairo, Egypt), conferring to the technique of Henry [23]. Urea concentration was determined according to the technique of Kaplan [24] using a reagent kit purchased from Diamond Diagnostics (Cairo, Egypt). Serum LDH activity was measured according to the method of Buhl and Jackson [25].

#### 2.6.2. RNA Isolation and qRT-PCR

Blood samples of all groups were used separately to extract total RNA via TRIzol Reagent purchased from Fermentas, Germany. Then cDNA synthesis was performed utilizing the High-Capacity cDNA Reverse Transcription Kit purchased from Invitrogen, Germany, according to the manufacturer’s directions.

Real-time PCR was carried out in a 20 μL system having 10 μL of 1× Sso-Fast Eva Green Supermix (Bio-Rad, Hercules, CA, USA), 2 μL of cDNA, 6 μL of RNase/DNase-free water, and 2 μL 500 nM of the primer pair sequences: NPR-1, F:5-AACAACGGCTCGGA CTGGAAGA-3 and R: 5-GGTAGATCCTGATGAATCGCGTG-3(NM001024628); ACE2, F: 5-TCCATTGGTCTTCTGTCACCCG-3 and R: 5-AGACCATCCACCTCC ACTTCTC-3 (NM021804.3) β–actin, F: 5-GGAACGGTGAAGGTGACAGCAG-3 and R-5-TGTGGACTTGGGAGAGGACTGG-3 (XM004268956.3); TLR2, F: 5-ATCCTCC AATCAGGCTTCTCT3 and R-5ACACCTCTG TAGGTCACT GTTG3 (NM00131878 9.2); TLR4, F:5-ATATTGACAGGAAACCCCATCCA-3, and R: AGAGAGATTGAGT AGGGGCATTT-3 (NM138554.5); β– actin, F-5AGGAACGGTGAAGGTGACAGCA G-3 and R-5TGTGGACTTGGGAGAGGACTGG-3 (XM00426 8956.3).

The cycling program was as follows: 30 s at 95 °C, and then 40 cycles of 5 s at 95 °C and 10 s at 60 °C. For every reaction, a 65–95 °C ramp was conducted with a melting curve study. With each process, the threshold duration at which the fluorescent signal exceeded an arbitrarily defined threshold close to the middle of the log-linear amplification step was monitored, and the relative amount of mRNA was detected. The amplification data were analyzed by using the manufacturer’s programmer according to the Livak and Schmittgen methods [26], and the variables were normalized to β–actin. 

### 2.7. Statistical Analysis

The result values were accessible as mean ± standard error (SE). SPSS version 22 for Windows was used for data analysis (IBM Corp., New York, NY, USA) [27]. The statistical comparisons between groups were carried out by Duncan’s method for post-hoc analysis. By means of the Pearson correlation coefficients method, correlation analysis was estimated between different studied parameters. Values of *p* < 0.05, *p* < 0.01, and *p* < 0.001 were statistically significant at three levels whereas values of *p* > 0.05 were statistically non-significant.

## 3. Results

### 3.1. Demographic Data and Laboratory Findings

Table 1 shows that the age was insignificant (*p* > 0.05) in moderate and severe cases compared to healthy controls. Male patients in severe cases were more than female patients. On the other hand, BMI was insignificant (*p* > 0.05) in both infected groups compared to healthy control. 

Table 1 also shows the levels of the kidney and heart function biomarkers in all groups. Serum creatinine level was significantly higher in severe cases (*p <* 0.001) than in moderate and control cases. No significant difference was found between moderate and control cases. Serum urea level was significantly higher in both moderate (*p* < 0.01) and severe (*p* < 0.001) cases in comparison with healthy control; it was significantly higher (*p* < 0.001) in severe patients than moderate ones. The levels of serum high-sensitivity C-reactive protein (hsCRP) (*p* < 0.001), CK-MB (*p* < 0.001), and LDH (*p* < 0.01, *p* < 0.001) were significantly increased among the severe group and moderate group compared to healthy control. Serum troponin I was significant (*p* < 0.01) only in the severe group compared to healthy control. Serum (hsCRP) (*p* < 0.001) and levels of lactate dehydrogenase (LDH) (*p* < 0.001) were significantly higher among the severe group compared to the moderate group.

### 3.2. The mRNA Expression Levels of TLR2 and TLR4 in Moderate and Severe COVID-19 Patients

The TLR2 and TLR4 mRNA expression levels were significantly higher (*p* < 0.01) in moderate and severe COVID-19 groups than in the healthy control group. When the severe COVID-19 group was compared with the moderate group, there was no significant change (*p* > 0.05) (Figure 1).

### 3.3. The mRNA Expression Levels of ACE2 and NPP-2 in Moderate and Severe COVID-19 Patients

The ACE2 and NPP-2 mRNA expression levels were significantly higher (*p* < 0.001) in moderate and severe COVID-19 groups than in the healthy control group. There were no significant differences (*p* > 0.05) when severe COVID-19 patients were compared with the moderate COVID-19 group (Figure 2).

### 3.4. Correlation between TLR2, TLR4, Renal, and Cardiac Biomarkers in the Moderate COVID-19 Patients

The data in Figure 3 show a significant positive correlation between serum urea and TLR2 mRNA expression (*p* = 0.000) and a positive correlation between serum CK-MB activity and Troponin I level with the mRNA expression of TLR2 (*p* = 0.000) and TLR4 (*p* = 0.000) in the moderate COVID-19 group.

### 3.5. Correlation between TLR2, TLR4, Renal, and Cardiac Biomarkers in the Severe COVID-19 Patients

Figure 4 depicts positive significant correlations of creatinine with TLR2 (A, *p* = 0.000), creatinine with TLR4 (B, *p* < 0.01), urea with TLR2 (C, *p* = 0.000), and urea with TLR4 (D, *p* = 0.000) in severe COVID-19 patients.

Figure 4 also shows a positive significant correlation between serum CK-MB activity with the mRNA expression of TLR2 (*p* = 0.000) and TLR4 (*p* < 0.01) in the severe COVID-19 group. The serum troponin levels have a positive significant correlation with TL4 mRNA expression (*p <* 0.01) in severe cases of the disease.

### 3.6. Correlation between ACE2 and TLR2 and TLR4 mRNA Expression in the Moderate COVID-19 Patients

The data in Figure 5 show a positive correlation between ACE2 mRNA expression and TLR2 (*p* = 0.000) and TLR4 (*p* = 0.000) mRNA expression in the moderate COVID-19 group.

### 3.7. Correlation between ACE2 and TLR2 and TLR4 mRNA Expression in the Severe COVID-19 Patients

ACE2 mRNA expression level exhibited a significant positive (*p* = 0.000) correlation with both TLR2 and TLR4 mRNA expressions in the severe COVID-19 group (Figure 6).

### 3.8. Correlation between NRP-1 and TLR2 and TLR4 mRNA Expression in the Moderate COVID-19 Patients

NRP-1 mRNA expression level had a significant positive (*p* = 0.000) correlation with both TLR2 and TLR4 mRNA expressions in moderate COVID-19 patients (Figure 7).

### 3.9. Correlation between NRP-2 and TLR2 and TLR4 mRNA Expression in the Severe COVID-19 Patients

ACE2 mRNA expressions were positively (*p* = 0.000) correlated with TLR2 and TLR4 mRNA expressions in the severe group (Figure 8).

### 3.10. Correlation between Renal and Cardiac Biomarkers with ACE2 and NRP-1 mRNA Expressions in Moderate COVID-19 Patients

Table 2 shows the correlation between both receptors with kidney and heart function biomarkers in the moderate group of COVID-19 patients. Serum creatinine showed an insignificant negative correlation with ACE2 and NRP-1 in moderate cases (*p* > 0.05). Serum urea was significantly correlated (*p* = 0.01) with the increase in NRP-1. Serum CK-MB (*p* = 0.000) and Troponin I (*p* = 0.009) concentration had significant positive correlations with NRP-1 and ACE2 mRNA expression levels. LDH was positively correlated only with NRP-1 (*p* < 0.05).

### 3.11. Correlation between Renal and Cardiac Biomarkers with ACE2 and NRP-1 in Severe COVID-19 Patients

Table 3 shows the correlation between mRNA of both receptors with kidney and heart functions in serum in the severe group of COVID-19 patients. Serum creatinine and blood urea, as renal biomarkers, were positively correlated with ACE2 and NRP-1 mRNA expression levels in the severe group (*p* < 0.05 and *p* < 0.01, respectively). On the other hand, cardiac enzymes, including CK-MB and LDH activities, have significant positive correlations with ACE2 and NRP-1 mRNA expression levels (*p* = 0.000). Troponin I showed a significant positive correlation with ACE2 mRNA expression levels only.

## 4. Discussion

COVID-19 is a global pandemic. From the start of its spread in December 2019, this viral infection continued to spread internationally. Efforts were instantly underway to stop this virus invasion and guard the world.

To progress an effective treatment and vaccine for COVID-19 infection, decoding the precise immune response to infection was an essential step. In this case, recovery and disease severity were linked to appropriate immune responses and impaired immune reactions, respectively. Compared to other routine laboratory tests, it was reported that CRP level was significantly associated with COVID-19 severity [28,29]. Consequently, CRP on admission is considered a simple and independent factor that can be helpful for early detection of COVID-19 severity and thereby facilitates the guidance of treatment decisions [30]. The current investigation lends credence to this finding because the CRP levels in moderate and severe COVID-19 individuals were shown to have increased by 31 and 38 times, respectively. The severity of the COVID-19 disease may deleteriously affect the normal functions of many organs, including the kidney and heart.

Our present data has revealed an increase in serum creatinine and urea levels in COVID-19 patients compared to healthy individuals. In agreement with our results, Cheng et al. have shown an elevation in both serum creatinine and urea levels in COVID-19 patients and near 5% of their prospective study. In China, patients were diagnosed with acute kidney injury during hospitalization [31]. Many authors of various publications [32,33,34,35] reported that kidney dysfunction—mainly acute kidney injury—occurs in cases of COVID-19 within 3 weeks of the beginning of symptoms; nevertheless, kidney complications are linked to a higher death rate. They concluded that COVID-19-induced organ impairment is mostly facilitated by cytokine storms and that strategies to diminish or eradicate inflammatory cytokines would be effective in avoiding cytokine-induced organ injury, aiming to reduce IL1, IL6, TNF-α, and INF-γ levels [32,33,34,35]. Kidney injury induced by the SARS-CoV-2 virus is probably related to several factors. The virus can infect renal podocytes and cells of the proximal tubules. It can also induce disruptive glomerulopathy, leakage of protein in Bowman’s capsule, acute tubular necrosis, and mitochondrial damage based on the angiotensin-converting enzyme 2 (ACE2) pathway [36]. Acute kidney injury can result from dysregulation of the immune response, such as cytokine storms and macrophage activation syndrome with lymphopenia [37]. The development of cytokine storms after viral infection, directly and indirectly, affects the kidneys and can induce sepsis, shock, hypoxia, and rhabdomyolysis. Among the other probable causes of kidney injury induced by COVID-19 is the organ interactions between the pulmonary, cardiac, and renal tissues [38].

The current study revealed that the serum levels of cardiac enzymes, including CK-MB and LDH, were significantly increased in both moderate and severe COVID-19 patients compared to healthy subjects; serum LHD activity was significantly higher in severe than moderate patients. The serum troponin I was significantly increased only in severe cases. These results are in concordance with those of previous publications [39,40,41,42], which recommended that patients check cardiac markers at admission, which helps medical staff predict the severity of patients in the later stage, and these cardiac biomarkers in serum could act as prognostic biomarkers in COVID-19 patients. As stated by Li et al. [40] and Sultan et al. [43], the mechanisms of actions of COVID-19 to induce heart injury include specific binding to functional receptors on cardiomyocytes and immune-mediated myocardial injury. This postulation was supported by the results of the present study, which revealed a positive significant correlation between serum activity of CK-MB, a potent biomarker of cardiomyocytes’ injury, and ACE2, NRP-1, TLR2, and TLR4 expression levels.

The present study detected a highly significant upsurge of ACE2 and NRP-1 mRNA expressions in COVID-19 patients. This finding was consistent with the results of Imig and Ryan [38], Cantuti-Castelvetri et al. [10], Daly et al. [44], and Freeman and Swartz [45], who reported that one of the major receptors for SARS-CoV-2 is ACE2, which is broadly dispersed in the lung, intestinal epithelium, liver, heart, vascular endothelium, testis, and kidney cells.

The fact that ACE2 is expressed in respiratory and olfactory epithelial cells at very low protein levels increases the likelihood that other factors are needed to promote virus-host cell communications in the cells that express low ACE2. NRP-1 may be considered an ACE2 enhancer by facilitating the interface of the virus and ACE2. In addition, TLRs are associated with acute kidney injury, which correlates with the severity of renal disease and inflammatory markers [10,38,44,45]. Moreover, Rivero et al. and Choudhury and Mukherjee concluded that TLRs (2 and 4) can encourage the expression of chemokine in epithelial cells of renal tubules [46,47]. Viruses interrelate with precise receptors to enter target cells. SARS-CoV-2 interacts with ACE2, which is broadly distributed in the lung, intestinal, liver, heart, vascular endothelium, testis, and kidney cells [48].

The spike protein S1 subunit of SARS-CoV-2 has a c-terminal domain that guarantees a very high attraction for the ACE2 receptor [49]. This decreases the ACE2 expression on the cell surface and upsurges inflammation leading to tissue destruction [50]. ACE2 plays an anti-inflammatory role by converting angiotensin II into angiotensin (1–7) [51] and reducing vaso-permeability, edema, and pulmonary neutrophil infiltration.

Ziegler et al. recommended that SARS-CoV-2 may raise ACE2 expression and further increase infection [52]. The spike protein of SARS-CoV-2 comprises a cleavage site for the protease furin, which is lacking in SARS-CoV. Cantuti Castelvetri et al. presented that NRP-1, which is identified to bind to furin-cleaving substrates, enhances SARS-CoV-2 infectivity [10]. 

Daly et al. reported that the furin-cleaving S1 fragment of the spike protein directly combines with cell surface NRP-1, and the blocking of this contact with small molecule inhibitors or monoclonal antibodies will reduce viral invasion in cell culture [44]. Viral invasion initiates inflammation and stimulation of specialized antigen-presenting cells (APCs) that make the viral peptides presentable to T cells (CD4 and CD8) and stimulates B cells directly. Inflammation is dependent on the first line of the innate immune response [53]. As soon as SARS-CoV-2 enters the host cell, it triggers the pyrin domain containing 3 (NLRP3) inflammasome, one of the Nod-like receptor family. Once the viral RNA interacts with TLRs 3, 7, 8, and 9, downstream of the NF-κB pathway becomes activated, which augments the production of pro-inflammatory cytokines [54].

Our study also revealed that there were positive significant correlations between cardiac and renal functions, ACE2 and NRP-1. New onset of heart failure (HF) was observed in as much as a quarter of hospitalized COVID-19 patients and in as much as one-third of those admitted to the intensive care unit (ICU) [55,56], despite not having a history of HF. This could be due to the direct effect of the virus or the systemic inflammation of the heart. Severe acute myocarditis can be a manifestation of the infection resulting in cardiogenic shock, which can then result in multi-organ dysfunction syndrome (MODS) and death [57]. In our opinion, the positive significant correlation between the heart and kidney function biomarkers, which were generally more deteriorated in severe COVID-19 patients, and expression levels of receptors ACE2 and NRP-1, may nominate these receptors to be important therapeutic targets in COVID-19 treatment strategies.

In the current study, TLR2 and TLR4 showed significantly higher expression in the blood of moderate and severe COVID-19 patients than in healthy individuals. Their expressions were positively correlated with kidney and heart function biomarkers and ACE2 and NRP-1 mRNA expressions in moderate and severe COVID-19 patients. Thus, TLR2 and TLR4 could be possible targets to prevent the severity of COVID-19 and develop therapeutic strategies for the disease.

TLR2 and TLR4 play a central role in the pathogenesis of diverse heart disorders. An early report showed that TLR2 and TLR4 were expressed in cardiomyocytes, a participant in responses of these cells to oxidative stress, and was a major contributor to the pathogenesis of cardiac dysfunction. Moreover, increased expression and signaling by TLR2 have been found to contribute to the activation of innate immunity in injured myocardium, indicating that TLR2 can promote myocardial inflammation in HF; however, another study indicated that TLR2 expressions in patients with chronic HF are similar compared with that in the control group [58]. TLR2 and TLR4 are crucial due to their exciting capability to identify different molecular forms of attacking pathogens [59]. Subsequently, the inflammation twitches, and the activated immune cells explode the release of a large number of cytokines. Collectively, these mechanisms participate in augmenting the inflammatory response that plays a major role in the pathogenesis of COVID-19 [60]. The “cytokine storm syndrome” is believed to be a major underlying factor in the immune-pathogenicity of COVID-19. This storm is responsible for the start of tissue damage, hyper-inflammation, and even mortality [61]. It was activated by the hyperactivation of immune cells leading to a boom of cytokine release. The overexpression of induced IL-10, macrophage IL-1 α, hepatocyte growth factor, IFN-γ, IL-3, and were highly related to the severity of the disease [62]. Freeman and Swartz concluded that SARS-CoV-2 is typified by strong and fast stimulation of the innate immune defense, involving triggering of the NLRP3 pathway and the release of the IL-6 and IL-1 [45]. Blanco-Melo et al. verified that SARS-CoV-2 infection of the epithelial cells of human bronchi gave rise to the expression of numerous cytokines and chemokines such as TNF-α, IL-1β, and IL-6 [62]. In addition, our present data showed elevated serum CK-MB and LDH activities in moderate and severe COVID-19 patients and an elevated serum troponin I level in severe COVID cases when compared with the control group. COVID-19 has been shown to interrelate and disturb the cardiovascular system, primarily via the ACE2 receptor, resulting in myocardial damage, cardiac damage, and endothelial dysfunction [63]. The interface of SARS-CoV-2 with ACE2 can give rise to changes in the ACE2 pathways, causing acute cardiac injury. A few studies propose that SARS-CoV-2 can result in viral myocarditis by direct infection of the myocardium. In many cases, myocardial damage seems to be triggered by increased cardiac metabolic demand, which accompanies systemic infections and persistent severe pneumonia-induced hypoxia [63,64]. Moreover, the release of high levels of interleukins 2, 10, 6, and 8, and TNF-α can harm several tissues, such as vascular endothelium and cardiac myocytes [61]. Cytokine storms may be associated with the dangerousness of the disease [65]. In a series of situations that affect the kidneys as well as the heart, dysfunction of one organ can encourage dysfunction of the other organ. Chronic or acute systemic conditions can impair the function of these two organs [66]. COVID-19-induced acute kidney injury may be related to the crosstalk between the cardiovascular system and the kidney. COVID-19 sets off myocarditis to weaken the cardiac output and affects end-organ perfusion. Additionally, the associated proper ventricular disorder produces diastolic disorder and venous congestion that transmit back to the kidney and, in addition, compromise its blood supply via growing kidney congestion [67]. In addition, acute viral myocarditis, together with cytokine cardiomyopathy, can induce hypotension, renal venous congestion, and decreased renal blood flow, which can decrease the glomerular filtration rate [68].

## 5. Conclusions

Our study concluded that impaired renal and cardiac biomarkers might be attributed to increased expression levels of TLR2, TLR4, ACE2, and NRP-1 mRNA in both moderate and severe COVID-19 patients. Thus, it is critical to recognize the molecular mechanisms and focus on the vital molecules involved in the pathogenicity of the diseases to design novel drugs to control and prevent the disease. By blocking the virus access pathways, including the viral receptors, and regulating immune responses, we can reduce the multi-organ dysfunction induced by COVID-19.

## Figures and Tables

**Figure 1 vaccines-10-01106-f001:**
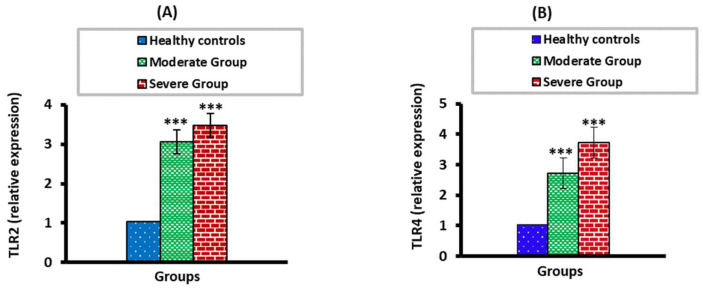
The mRNA expression levels of TLR2 (**A**) and TLR4 (**B**) in moderate and severe COVID-19 groups compared to healthy control. *** *p* < 0.001.

**Figure 2 vaccines-10-01106-f002:**
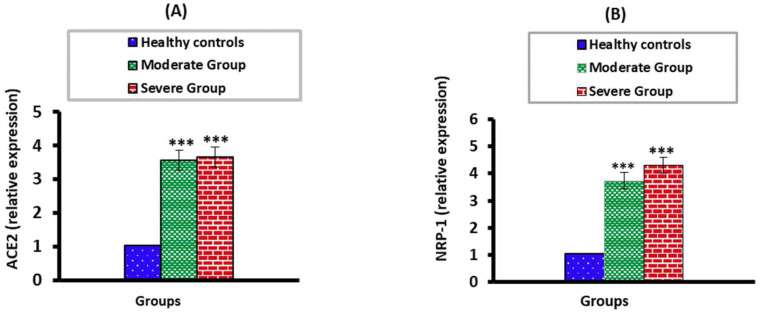
The mRNA expression levels of ACE2 (**A**) and NRP-2 (**B**) in moderate and severe groups compared to healthy controls. *** *p* < 0.001.

**Figure 3 vaccines-10-01106-f003:**
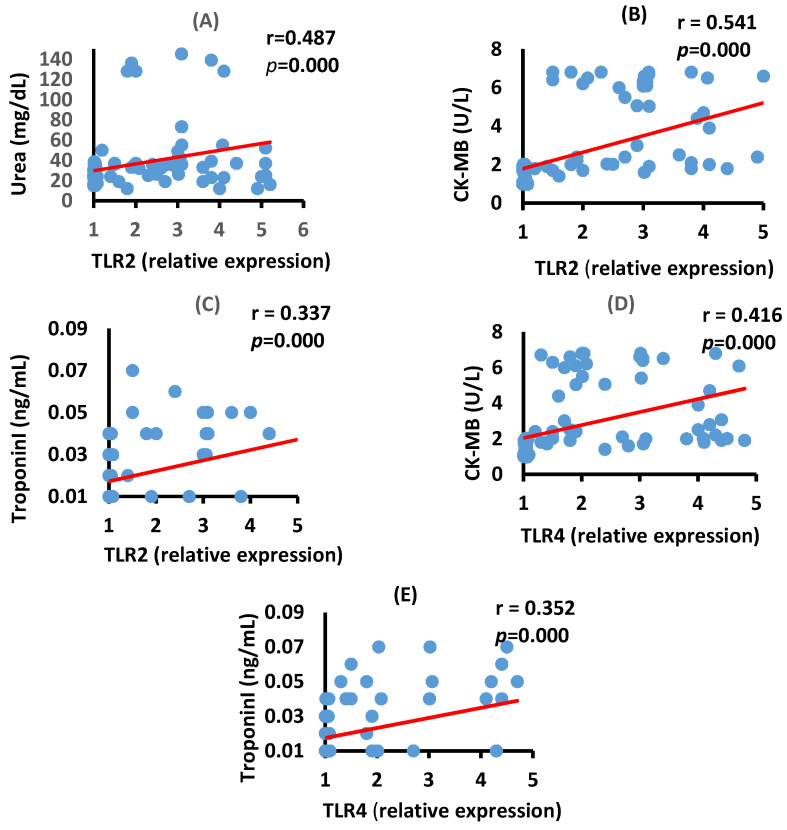
Correlation between kidney and heart function biomarkers in serum and TLR2 and TLR4 mRNA expression in moderate COVID-19 patients. Correlation between urea and TLR2 (**A**), *p* = 0.000, CK-MB and TLR2 (**B**), *p* = 0.000, Troponin I and TRL2 (**C**), *p* = 0.000, CK-MB and TLR4 (**D**), *p* = 0.000) and between Troponin I and TLR4 (**E**), *p* = 0.000.

**Figure 4 vaccines-10-01106-f004:**
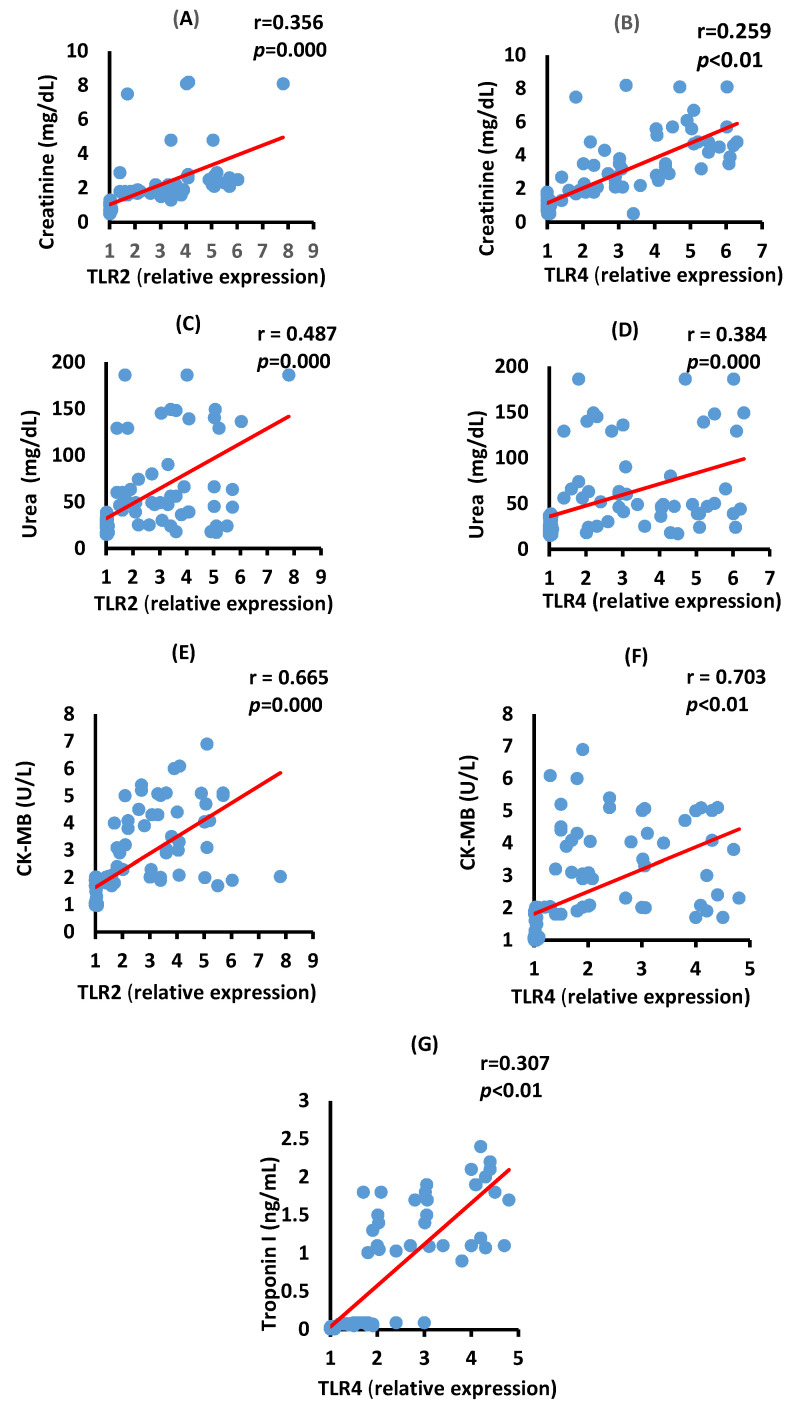
Correlation between kidney and heart function biomarkers in serum and TLR2 and TLR4 mRNA expressions in severe COVID-19 patients. Correlations between creatinine with TLR2 (**A**), *p* = 0.000), creatinine with TLR4 (**B**), *p* < 0.01), urea with TLR2 ((**C**), *p* = 0.000), urea with TLR4 ((**D**), *p* = 0.000), CK-MB with TLR2 (**E**), *p* = 0.000), TLR4 (**F**), *p* < 0.01), and Troponin I level with TLR4 (**G**), *p* < 0.01).

**Figure 5 vaccines-10-01106-f005:**
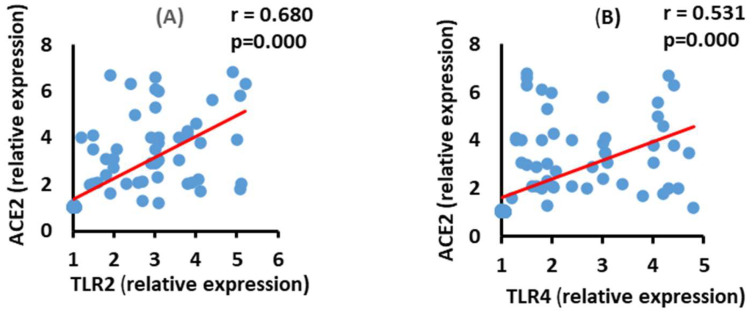
Correlations between ACE2 with TLR2 (**A**) (*p* = 0.000 and TLR4 (**B**), (*p* = 0.000) in moderate COVID-19 patients.

**Figure 6 vaccines-10-01106-f006:**
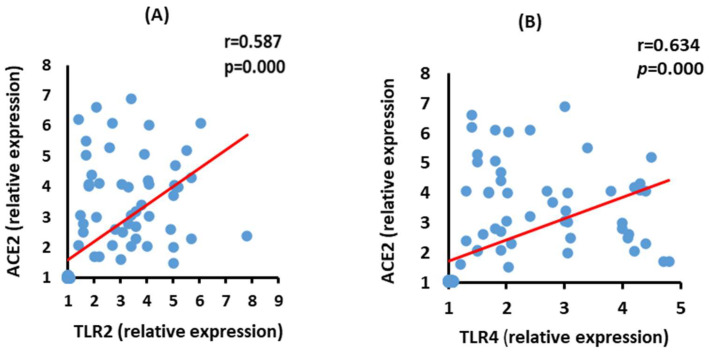
Correlations between ACE2 mRNA expression with TLR2 (**A**) (*p* = 0.000), and TLR4 (**B**) (*p* = 0.000) mRNA expression in severe COVID-19 patients.

**Figure 7 vaccines-10-01106-f007:**
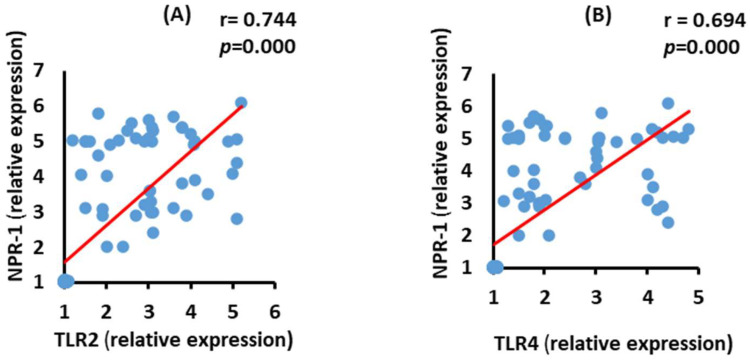
Correlations between NRP-1 mRNA expression with TLR2 (**A**) (*p* = 0.000), and TLR4 (**B**) (*p* = 0.000) mRNA expression in moderate COVID-19 patients.

**Figure 8 vaccines-10-01106-f008:**
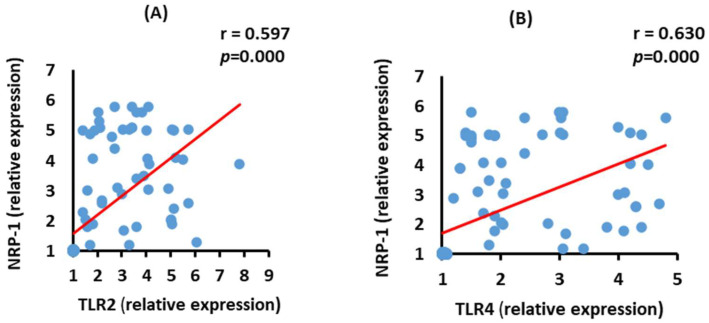
Correlations between NRP-1 mRNA expression with TLR2 (**A**) (*p* = 0.000) and TLR4 (**B**) (*p* = 0.000) mRNA expression in severe COVID-19 patients.

**Table 1 vaccines-10-01106-t001:** Demographic and laboratory findings of severe and moderate COVID-19.

	Healthy Controls	Moderate Patients	Severe Patients
	(*n* = 50)	(*n* = 50)	(*n* = 50)
Age (Year)	52.12 ± 1.22	58.24 ± 1.24	57.36 ± 1.43
Gender, no. (%)			
Male	25 (50%)	25 (50%)	29 (58%)
Female	25 (50%)	25 (50%)	21 (42%)
BMI (kg/m^2^)	24.25 ± 0.35	24.73 ± 0.48	25.03 ± 0.52
CRP (mg/dL)	1.97 ± 0.19	61.90 ± 6.05 ^+^ ***	73.92 ± 3.42 ***
Creatinine (mg/dL)	0.89 ± 0.03	0.85 ± 0.06 ^+++^	1.75 ± 0.30 **
Urea (mg/dL)	25.1 ± 0.93	48.68 ± 5.49 ^+++^ **	79.52 ± 9.14 ***
CK-MB (U/L)	1.29 ± 0.05	4.07 ± 0.29 ***	3.55 ± 0.20 ***
LDH (U/L)	261.6 ± 3.00	339.24 ± 25.96 ^++^ **	418.7 ± 26.12 ***
Troponin I (ng/mL)	0.02 ± 0.002	0.03 ± 0.003	0.04 ± 0.01 ***

Data are presented as mean +/− SE. BMI: body mass index, CRP: C-reactive protein, CK-MB: creatine kinase–myocardial Band, LDH: lactate dehydrogenase. Data are significantly different at ** *p* < 0.01 level, *** *p* < 0.001 level versus healthy controls, and ^+^ *p* < 0.05, ^++^ *p* < 0.01, and ^+++^ *p* < 0.001 versus severe group.

**Table 2 vaccines-10-01106-t002:** Correlation between renal and cardiac function biomarkers in serum with ACE2 and NRP-1 mRNA expressions in moderate COVID-19 patients.

	Renal Biomarkers				Cardiac Biomarkers					
	Creatinine		Urea		CK-MB		LDH		Troponin I	
	r	*p*	r	*p*	r	*p*	r	*p*	r	*p*
**ACE2**	−0.113	0.262	0.193	0.054	0.543 ***	0.000	0.023	0.823	0.260 **	0.009
**NRP-1**	−0.018	0.86	0.334 **	0.001	0.653 ***	0.000	0.198 *	0.048	0.261 **	0.009

Data are presented as mean +/− SE. Correlations were considered significant at three levels, * *p* < 0.05, ** *p* < 0.01, and *** *p* < 0.001.

**Table 3 vaccines-10-01106-t003:** Correlation between renal and cardiac function biomarkers in serum with ACE2 and NRP-1 mRNA expressions in severe COVID-19 patients.

	Renal Biomarkers				Cardiac Biomarkers					
	Creatinine		Urea		CK-MB		LDH		Troponin I	
	r	*p*	r	*p*	r	*p*	r	*p*	r	*p*
**ACE2**	0.204 *	0.041	0.438 ***	0.000	0.586 ***	0.000	0.557 ***	0.000	0.147	0.144
**NRP-1**	0.300 **	0.002	0.443 ***	0.000	0.624 ***	0.000	0.328 **	0.001	0.301 **	0.002

Data are presented as mean +/− SE. Correlations were considered significant at three levels, * *p* < 0.05, ** *p* < 0.01, and *** *p* < 0.001.

## Data Availability

The data are contained within the article.

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
