# Peer review of "Correlations between Kidney and Heart Function Bioindicators and the Expressions of Toll-Like, ACE2, and NRP-1 Receptors in COVID-19"

_vaccines, 2022, doi:10.3390/vaccines10071106_

Round 1

Reviewer 1 Report

The author detected the concentration of serum creatinine, blood urea, CKMB, LDH and Troponin I and the gene expression of TLR2, TLR4 ACE2 and NRP-1 in the blood samples from COVID-19 patients and healthy control, and found that CKMB activity, ACE2, and NRP-1 mRNA expression levels were positively correlated to both TLR2 and TLR4 expression levels. The innate immune markers TLR2 and TLR4 can be important since they correlate with the severity of illness in COVID-19.

1.       The title and conclusion may not be supported completely by their data, since the authors only showed CKMB activity, ACE2, and NRP-1 mRNA expression levels were positively correlated to both TLR2 and TLR4 expression levels. No data supported the correlation between ACE2, or NRP-1 mRNA expression levels and renal biomarkers and cardiac enzymes. Or the author should analysis the correlation between ACE2, or NRP-1 mRNA expression levels and renal biomarkers and cardiac enzymes.

2.       It is better to have subtitle in results, and organize the results, not just showed the results of table and figures.

3.       The figure legends of figures 3, 4, 5, 6, 7 8 showed correlation was significant ** at the 0.01 level,*** at the 0.001 level. However, no symbol ** or ,*** was found in these figures.    

4.       There was no discussion about the correlation between CKMB activity, ACE2, and NRP-1 mRNA expression levels and TLR2 and TLR4 expression levels in discussion section.

Author Response

Reviewer 1

Comments and Suggestions for Authors

The author detected the concentration of serum creatinine, blood urea, CKMB, LDH and Troponin I and the gene expression of TLR2, TLR4 ACE2 and NRP-1 in the blood samples from COVID-19 patients and healthy control, and found that CKMB activity, ACE2, and NRP-1 mRNA expression levels were positively correlated to both TLR2 and TLR4 expression levels. The innate immune markers TLR2 and TLR4 can be important since they correlate with the severity of illness in COVID-19.

  1. The title and conclusion may not be supported completely by their data, since the authors only showed CKMB activity, ACE2, and NRP-1 mRNA expression levels were positively correlated to both TLR2 and TLR4 expression levels. No data supported the correlation between ACE2, or NRP-1 mRNA expression levels and renal biomarkers and cardiac enzymes. Or the author should analysis the correlation between ACE2, or NRP-1 mRNA expression levels and renal biomarkers and cardiac enzymes.

Author response: Thank you for your comment. correlation between ACE2, or NRP-1 mRNA expression levels and renal biomarkers and cardiac enzymes in tables 2 and 3. (Pages 11 and 12)

  1. It is better to have subtitle in results, and organize the results, not just showed the results of table and figures.

Author response: Thank you for your comment. The results were organized. Subtitles were added for each section in results. Below each subtitle, the descriptions of related results were precisely added.  (Pages 5-12)

  1. The figure legends of figures 3, 4, 5, 6, 7 8 showed correlation was significant ** at the 0.01 level,*** at the 0.001 level. However, no symbol ** or ,*** was found in these figures.

Author response: Thank you for your comment. The descriptions of symbols and p values were added in footnotes below each table and figure    

  1. There was no discussion about the correlation between CKMB activity, ACE2, and NRP-1 mRNA expression levels and TLR2 and TLR4 expression levels in discussion section.

Author response: Thank you for your comment. The discussion of correlations between CK-MB activity, ACE2, and NRP-1 mRNA expression levels and TLR2 and TLR4 expression levels were added in discussion section. The added parts in discussion were marked red and highlighted with yellow color (Discussion Pages 12-15).

Reviewer 2 Report

The manuscript entitled "Correlations between kidney and heart function bioindicators and the expressions of Toll-Like, ACE2, and NRP-1 receptors in COVID-19"  reports that impaired renal and cardiac enzymes may be attributed to increased expression of TLR2, TLR4, ACE2,  and NRP1 mRNA in moderate and severe cases of COVID-19.

The data are very interesting and I am strongly inclined to recommend it for publication on "Vaccines", although i have some questions:

1. in the Introduction, the authors should describe the function of analyzed receptors in more details.

2. There are some typos in the text.

Author Response

Reviewer 2 Comments and Suggestions for Authors

The manuscript entitled "Correlations between kidney and heart function bioindicators and the expressions of Toll-Like, ACE2, and NRP-1 receptors in COVID-19"  reports that impaired renal and cardiac enzymes may be attributed to increased expression of TLR2, TLR4, ACE2,  and NRP1 mRNA in moderate and severe cases of COVID-19.

The data are very interesting and I am strongly inclined to recommend it for publication on "Vaccines", although i have some questions:

  1. in the Introduction, the authors should describe the function of analyzed receptors in more details.

Author response: Thank you for your comment. More details were added in the introduction about the analyzed receptors. The added parts were marked red and highlighted yellow.

  1. There are some typos in the text.

Author response: Thank you for your comment. The manuscript was double revised for corrections of typing and grammar errors

Reviewer 3 Report

The work done by sultan etal entitled:"Correlations between kidney and heart function bioindicators 2 and the expressions of Toll-Like, ACE2, and NRP-1 receptors in 3 COVID-19" need some modifications before publication.

In the introduction clearly explain the research question.

in the introduction, no information mentioned about the heart function normally

in methods section, explain in details the real time PCR parameters

Author Response

Reviewer 3 Comments and Suggestions for Authors

The work done by sultan et al entitled:"Correlations between kidney and heart function bioindicators 2 and the expressions of Toll-Like, ACE2, and NRP-1 receptors in 3 COVID-19" need some modifications before publication.

In the introduction clearly explain the research question.

Author response: Thank you for your comment. The introduction was revised to explain the research question. More details were added in the introduction about the analyzed receptors.

in the introduction, no information mentioned about the heart function normally

Author response: Thank you for your comment. Information about the heart function and biomarkers detected in serum to assess the heart function was added in introduction in page 3. The added text was marked in red color and highlighted yellow.

in methods section, explain in details the real time PCR parameters.

Author response: Thank you for your comment. Thank you. The real time PCR was described in more details in page 5. The text of this part was marked red and highlighted yellow.

Round 2

Reviewer 1 Report

The correlation between renal and cardiac function biomarkers in serum with ACE2 and NRP-1 mRNA expressions in COVID-19 patients should be described in the results of the abstract.  

Author Response

Reviewer 1 comments

The correlation between renal and cardiac function biomarkers in serum with ACE2 and NRP-1 mRNA expressions in COVID-19 patients should be described in the results of the abstract. 

Author response: Thank you for your comment.  The correlations between renal and cardiac function biomarkers in serum with ACE2 and NRP-1 mRNA expressions in COVID-19 patients (in moderate and severe groups) were described in the results of the abstract. The added text was marked red and highlighted yellow.